# The Effects of Artificial Diets Containing Free Amino Acids Versus Intact Proteins on Biomarkers of Nutrition and Deformed Wing Virus Levels in the Honey Bee

**DOI:** 10.3390/insects16040375

**Published:** 2025-04-02

**Authors:** José Carlos Tapia-Rivera, José María Tapia-González, Mohamed Alburaki, Philene Chan, Rogelio Sánchez-Cordova, José Octavio Macías-Macías, Miguel Corona

**Affiliations:** 1Centro de Investigaciones en Abejas, Centro Universitario del Sur, Universidad de Guadalajara, Guadalajara 44100, Mexico; jose.tapia@cusur.udg.mx (J.C.T.-R.); joset@cusur.udg.mx (J.M.T.-G.); joseoc@cusur.udg.mx (J.O.M.-M.); 2Honey Bee Research Laboratory, United States Department of Agriculture, Beltsville, MD 20705, USA; mohamed.alburaki@usda.gov (M.A.); philene.chan@usda.gov (P.C.); 3Industrial Rosanco S.A de C.V, Orizaba Veracruz 94390, Mexico; rosanco2001@yahoo.com.mx

**Keywords:** pollen substitutes, nutrition, vitellogenin, deformed wing virus, *Apis mellifera*

## Abstract

This study compared the effects of pollen substitutes containing intact proteins with those composed of free amino acids on honey bees’ nutritional status and health in experimental caged bees. We compared the mRNA levels of vitellogenin (*vg*) and major royal jelly protein 1 (*mrjp1*)—two nutritionally regulated genes—as nutrition biomarkers. We also measured the deformed wing virus (DWV) levels as an indicator of honey bee health. We did not find significant differences in *vg* levels between bees fed intact protein and free amino acid diets. However, *mrjp1* mRNA levels were higher in bees fed the free amino acid diets. These results indicate that varying amino acid availability affects the expression of these nutritionally regulated genes differently. These genes could be utilized together to assess the nutritional value of different pollen substitutes. Our study also reveals that although pollen substitutes with free amino acids induce high expression levels of nutritional biomarkers, they also cause a faster increase in DWV levels and mortality.

## 1. Introduction

Around 70% of the plant species used for food production worldwide rely on animal pollination primarily carried out by bees [1,2]. However, honey bee populations significantly decreased in the United States and other countries in recent years [3,4,5,6]. On an economic level, the decline in honey bee populations impacts honey production and diminishes the yield of fruit and vegetable crops that rely on bee pollination for fertilization [2]. Several factors have been proposed to explain the loss of honey bee colonies, including the impact of the ectoparasite mite *Varroa destructor* [7] and its synergistic interactions with highly pathogenic viruses, such as the deformed wing virus (DWV) [8]. Nutritional stress is another factor proposed to contribute to honey bee colony loss. The effect of this factor on colony losses has been demonstrated by studies showing that bees in areas with intensive agriculture have lower levels of physiological biomarkers of nutrition, including storage proteins and lipids, compared with bees maintained in areas of less cultivation [9,10,11,12].

Nutritional resources for honey bees primarily include nectar and pollen, which provide carbohydrates, proteins, lipids, vitamins, and minerals [13,14,15]. Pollen is bees’ main source of protein and lipids, and its consumption is essential for their growth and development [13,14,15,16,17]. However, colonies surrounded by monocultures experience pollen shortages before and after flowering [18]. In addition, since the pollen of some plant species is deficient in certain amino acids [19,20,21,22,23,24] and fatty acids [25], bees that consume pollen from a limited number of plants are less likely to have balanced nutrition. Hence, increased evidence shows that reduced floral diversity in regions with intensive agriculture decreased both the quantity and quality of pollen collected by bees [10,11,18,26]. The amount of protein in pollen from various plants varies [19,27,28]. However, the nutritional value of a particular pollen (“pollen quality”) is mainly determined by its composition of amino acids. Bees, similar to other animals, cannot produce the ten amino acids essential for their nutrition. These nutritionally essential amino acids must be present in the correct proportions (“de Groot ratio”) in the bees’ diet for proper utilization during protein synthesis [29,30]. Pollen also contains several types of lipids that are critical for honey bee physiology [25,31]. The fatty acids are precursors of complex lipids, including phospholipids (e.g., phosphatidylcholine), with vital structural functions in cell membranes [31,32]. Two polyunsaturated fatty acids (PUFAs) are considered essential in insects, including honey bees: alpha-linolenic acid (omega-3) and linoleic acid (omega-6) [33], which the bee must consume in a 1:1 ratio for optimal cognitive performance [34]. 

Developing artificial diets for bees is needed to address the pollen shortage during cold and dry seasons [35,36,37]. Nutritional supplements for honey bees encompass a wide range of products and strategies [15]. The potential benefits of nutritional supplementation for honey bees include supporting brood rearing and increasing honey bee colonies’ overall productivity [38]. Pollen substitutes can be categorized based on several criteria, including their physical state and constitution. These artificial diets typically contain amino acids in two forms: intact proteins (IPs), where amino acids are covalently linked together, and free amino acids (FAAs), in which the amino acids remain separate. Solid diets often contain IPs from plants (e.g., soy), yeast, and microalgae [15,38,39]. On the other hand, FAA diets are typically prepared as liquid solutions [40]. They can be classified as containing total amino acids (TAA) or essential amino acid (EAA) blends based on their amino acid composition [41]. A TAA diet includes all twenty amino acids found in proteins, whereas an EAA diet consists of the ten essential amino acids the body cannot synthesize [29]. Solid diets are typically fed to bee colonies as patties placed inside the hive, while liquid diets are fed in top or internal feeders [38,42]. Different studies in mammals compared the effects of ingesting FAAs and protein hydrolysates versus IPs in terms of nutrient absorption, protein synthesis, and overall metabolic response [43,44,45,46]. FAAs and protein hydrolysates are quickly absorbed in the intestine, increasing plasma amino acid levels and muscle protein synthesis rates [43,44,45]. In contrast, IPs may have a more sustained effect on amino acid availability [45]. On the other hand, evidence from various studies on vertebrates suggests that consuming EAA alone or with IPs can enhance protein synthesis [46,47] and decrease protein degradation [48]. In honey bees, diets with high concentrations of IPs or EAAs have been associated with increased mortality, suggesting that the bees may be unable to metabolize these components at high levels [49,50]. An intact protein-to-carbohydrate (IP:C) ratio of 1:3 produced the highest ovarian activation in bees that were fed royal jelly, although higher IP:C ratios were associated with increased mortality [50]. On the other hand, the optimal EAA-to-carbohydrate (EAA:C) ratio varies with the bee’s age, ranging from 1:50 in nurse bees to 1:75 in foragers. However, higher EAA:C ratios also increased mortality [50]. These studies reveal that, contrary to common assumptions, pollen substitutes with high concentrations of proteins or amino acids may create a nutritional imbalance that is harmful to bee health.

Recent research on honey bee physiology revealed clusters of coregulated genes associated with honey bees’ physiological and behavioral development [51]. A group of highly expressed genes in nurses include nutritionally regulated transcripts encoding for vitellogenin (*vg*) [51,52] and the major royal jelly protein 1 (*mrjp1*) [51,53]. These findings on physiological research have practical implications for the development of nutrition biomarkers since pollen ingestion has been shown to trigger *vg* expression both in caged bees [26,54,55,56,57] and colony-level experiments [9,10]. Additionally, diets rich in IPs alone or supplemented with amino acids have also been found to induce *vg* expression [58,59], making this gene the most widely used molecular biomarker of nutrition in honey bees. On the other hand, while the expression of *mrjp1* is associated with the development of the hypopharyngeal glands—another indicator of honey bee health and nutritional status [60]—and is also significantly induced by pollen ingestion [51,57], it has not been used as a molecular biomarker of nutrition. In addition to studying the impact of pollen substitutes on honey bee nutrition, a growing concern in beekeeping involves investigating how nutritional supplementation affects the levels of prevalent pathogens, including DWV [61]. However, previous studies have shown conflicting results [39,51,55,60,61], highlighting the necessity for more studies with multiple factors to consider when evaluating the effect of nutrition on viral infections. 

This study aimed to assess the effects of pollen substitutes composed of IPs and/or FAAs on honey bees’ nutritional status and health, specifically in caged bees. To measure the effects of these diets, we observed bee survival and compared the levels of two molecular biomarkers of nutrition (*vg* and *mrjp1*) and DWV. We hypothesized that since FAAs do not need to be digested and are easily absorbed in the intestine, they would quickly affect the levels of nutritionally regulated genes encoding storage proteins. 

## 2. Materials and Methods

### 2.1. Caged Honey Bees 

Experiments were initiated using newly emerged bees from 20 colonies maintained during the summer of 2022 at the experimental fields of the Centro de Investigaciones en Abejas (CIABE) of the University of Guadalajara in Ciudad Guzman Jalisco, Mexico (19.7252, −103.4610, 1507 m above sea level). Bees were then transported to a laboratory and maintained under controlled conditions. Colonies were derived from populations of *Apis mellifera carnica* and headed by open-mated sister queens. *Varroa* mite infestation levels were measured two months before the experiment using the alcohol wash technique [62]. Colonies with *Varroa* infestation levels exceeding 5% were treated with a thymol-based product [Happy Varr^®^, Queretaro, Qro, Mexico]. One month before the experiment, *Varroa* mite infestation levels were measured again, showing levels not exceeding 1%. We selected two capped brood frames containing bees about to emerge. Each frame was placed in a wire mesh cage built with a 3 mm diameter mesh sieve, measuring 56 cm in length, 6 cm in width, and 29 cm in height. These caged frames were placed in the central part of two standard hives, with one frame in each hive, and left there for two days to allow the bees to emerge. This procedure promotes interactions between newly emerged bees (NEBs) and older colony mates, facilitating the transmission of microbiota and protein-rich secretions. After hatching, NEBs (≈1-day-old) were thoroughly mixed in a plastic container, and two hundred bees were randomly distributed per cage in triplicate. Additionally, a group of NEBs were frozen and collected as controls before providing any treatment. The remaining NEBs were maintained at 32 °C and 60% relative humidity. For an overview of the experiment, please refer to Figure 1.

### 2.2. Honey Bee Diets and Age Collections

Bees were provided with different diets across three cage replicates for each treatment, and their survival was monitored over a period of 45 days. The diets included three types of pollen substitutes, along with a control diet consisting of sugar syrup (sugar): 1. A liquid, FAAs diet with total amino acids (TAA). 2. A solid, IPs soy-based diet: Ultra bee^®^ (UBee) [Mann Lake Inc., Hackensack, MN, USA]: 3. A solid, IPs yeast-based diet, supplemented with TAA: Apitir^®^ [Técnica Industrial Rosanco S.A de C.V; Orizaba, Ver, Mexico]. 4. Sugar syrup 50% (sugar) control group. Apitir and UBee are proprietary diets available for commercial use. TAA is a diet developed at the USDA Bee Research Laboratory in Beltsville, MD, USA. This diet has been previously used in experiments to evaluate the effect of phytochemicals on bee pathogens [63] and stimulate egg laying in honey bee queens confined to small laboratory colonies [40]. We prepared the TAA diet using a blend of commercially available FAAs with 99.94% purity [NeutraBio^®^, Bolingbrook, IL, USA], in which the amounts of the EAA [29] were adjusted according to de Groot ratios [30]. EAA and non-essential amino acids were added at the same molarity. The final concentration of all twenty amino acids in the TAA diet was 22 mM, with an amino acid to carbohydrate ratio (AA:C) of 1:75. Lecithin, flaxseed oil, and safflower oil were included as sources of essential fatty acids (Edwards, International, Chicago, IL, USA). Diet composition was analyzed at the CUSUR bromatology laboratory, University of Guadalajara. The crude protein and amino acid content analysis was conducted using the Kjeldahl method to measure nitrogen, following a conversion factor [64]. The fat content was measured using the Soxhlet method [65] (Appendix A). To feed the bees, two gravity plastic feeders with a 5 mL capacity were placed on top of the cages. One feeder contained purified water, while the other contained 50% sugar syrup. The group that received FAAs was given the TAA diet instead of sugar syrup. The two groups fed IPs (Apitir, UBee) had sugar syrup added in a 1:1 ratio until the powder reached a paste consistency. Then, 2.5 g of the prepared patty was given per cage. Every two days, the remaining food from each diet was weighed to determine the accumulated consumption, and fresh food was provided. Age collections were performed when the bees were 1 week old (W1), 2 weeks old (W2), and 3 weeks old (W3). Three bees were collected per cage and time collection (*n* = 9) (Figure 1). Individual bees were placed in 2 mL plastic tubes containing 1 mL of RNA [29] ^®^ solution [Thermo Fisher Scientific Inc., Waltham, MA, USA]. Samples were stored at −80 °C until they were transported to the USDA Bee Research Laboratory (Beltsville, MD, USA) for molecular analyses. 

#### 2.2.1. Food Consumption and Honey Bee Survival

We measured the food consumed by weighing the feeders before placing them in the cages and then again after filling them with feed. The daily feed consumption was then calculated until the last bee perished. The final weight was adjusted based on the bees’ mortality rate per treatment and cage. We recorded mortality daily and calculated survival per treatment once the last bee died. Kaplan–Meier survival analysis was conducted, in which 1 represents the maximum probability of survival, and 0 the null probability of life during the days they were alive.

#### 2.2.2. Hemocytes Measurements

Thirty NEBs were sampled for hemocyte measurement at day 1. Subsequently, three bees were collected daily per nutritional group until they died. A small puncture was made in each bee’s second and third abdominal tergite using a sterile needle, and this area was gently compressed to obtain 4 μL of hemolymph. The hemolymph sample was collected with a micropipette and evenly spread on a microscope slide previously divided into 25 mm^2^ quadrants for easier hemocyte counting. The sample was air-dried at room temperature for approximately 20 min. Once dry, the smear was immersed five times for one second each in a fixative solution composed of 37% formaldehyde, distilled water, monobasic sodium phosphate, and dibasic sodium phosphate and allowed to air dry at room temperature for an hour. After drying, the samples were stained using the HYCEL^®^ Rapid Blood Stain Kit [Zapopan, Jal, Mexico] protocol. Each slide was dipped five times for 2 s each in Solution I (Eosin), then rinsed with distilled water to remove excess water. Next, the samples were immersed in Solution II (Methylene Blue) five times for 2 s. The slide was rinsed with distilled water and air dried at room temperature, resting at a 60° angle overnight. After staining, the samples were analyzed for hemocytes using an optical microscope at 100× magnification with oil immersion. The number of hemocytes per microliter was calculated using a modification of the method described by Koleoglu et al. (2018) with the following formula: (number of hemocytes counted per slide/16) × (3950)/4 [66]. The hemocyte count was not conducted in the group of bees fed with Apitir^®^ because of the limited number of surviving bees.

#### 2.2.3. RNA Extraction and Quantitative PCR of *vg*, *mrjp1* and DWV

RNA extractions were conducted using individual whole-body bees. Frozen samples stored in RNAlater solution were thawed on ice and transferred into 2 mL homogenization tubes containing 30 mg of ceramic beads (1.4–1.7 mm Yttria Stabilized Zirconia YSZ^®^), [MSE sup, Tucson, AZ, USA] and 1 mL of RLT buffer RNeasy^®^ kit [Qiagen, Germantown, MD, USA]. Bees were homogenized using a fast-prep-24 5G^®^ [MP Biomedicals, Irvine, CA, USA] sample preparation system at 6.0 m/s for 2 min, two times. Then, 900 μL of chloroform was added to each sample and mixed for 30 s. Half of the volume of the homogenized solution (900 μL) was then transferred to a new tube for further processing. The remaining sample volume was stored at −80 °C as a backup. RNA extraction was continued by adding 900 μL of chloroform to each sample, and then the manufacturer’s protocol was followed without further modifications. 

The *vg*, *mrjp1,* and DWV mRNA levels were analyzed using two-step RT-qPCR using as internal control the gene encoding for the ribosomal protein S5 (*rps5*). For each sample, cDNA was synthesized using 1 μg of the total RNA and Thermo Fisher reagents [Burlington, ON, Canada], including M-MLV reverse transcriptase (40U), RNase inhibitor (25U), random hexamers (2.5 μM), and dNTPs (0.8 mM) in a final reaction volume of 25 μL. The thermal profile for cDNA synthesis was as follows: 25 °C (10 min), 48 °C (45 min), and 70 °C (5 min). Each cDNA reaction was diluted by adding 100 μL buffer (10 mM Tris HCl pH 8.5). Transcription levels were quantified by qRT-PCR using a Life Technologies Vii7 system [Burlington, ON, Canada], with SYBR green reagents and a two-step thermal profile for amplification (95 °C 15 s, 60 °C 1 min). All qPCR assays were conducted in triplicates with reverse and forward primers, having a final concentration of 300 nM in a 10 μL reaction volume. Primer sets for amplifying the different targets are detailed in Appendix A [51,67,68]. Quantification of mRNA levels was performed by the ΔΔCT method, and relative values were calculated based on the differences (ΔCT) between the CT values of the focal gene/virus and the *rps5* control gene. ΔCT values exceeding two standard deviations (SD) from the mean were considered technical outliers and removed from the analysis. The effect of different diets on the mRNA levels of *vg*, *mrjp1*, and DWV was examined in NEBs and W1 workers. Subsequent measurements were only carried out on the diet groups with sufficient surviving bees for molecular analyses (*n* = 9). Bees fed with TAA were excluded from further analyses after 2 and 3 weeks due to the higher mortality observed during the first 2 weeks.

#### 2.2.4. Statistical Analysis

The survival rate data were analyzed using Kaplan–Meier analysis. Analyses of normality were conducted using the Shapiro–Wilk test. Non-normal data were analyzed using the Kruskal-Wallis test, followed by multiple comparison post hoc tests using RStudio (Version 2024.12.1 +563) [69]. The Mann–Whitney U test was used to analyze paired comparisons among treatments. 

## 3. Results

### 3.1. Bee Survival and Food Consumption

The survival curve reveals a clear contrast between treatments, with bees fed UBee having significantly higher survival rates than those fed other treatments (UBee vs. sugar *p* = 0.016, UBee vs. Apitir *p* = <0.001, and UBee vs. TAA *p* = 0.005) (Figure 2). On the other hand, while the bees fed TAA and sugar syrup had the lowest survival rate during the first ten days, the bees fed with Apitir experienced an accelerated mortality rate after this initial period. Our data also reveal distinct patterns in food, sugar syrup, and water consumption across different types of diets. Bees fed Apitir and TAA consumed the most, while those on the UBee diet consumed the least (Figure 3). There were significant differences in the consumption of diets containing FAAs (Apitir and TAA) and those of diets containing exclusively IPs (UBee) (*p* < 0.001). In contrast, bees fed UBee had significantly higher sugar syrup consumption than Apitir (*p* = 0.02). Additionally, our results indicate that bees fed Apitir treatment consumed the highest amount of water, followed by those fed with UBee (Apitir vs. UBee *p* = 0.007). 

### 3.2. Hemocyte Number

The number of hemocytes was high in newly emerged bees but declined in older bees from all the experimental groups (*p* < 0.001) (Figure 4). Bees that were fed with UBee showed a tendency to have more hemocytes compared to those fed with UBee and control sugar syrup, although this difference did not reach statistical significance.

### 3.3. Analyses of Gene Expression and Viral Levels

#### 3.3.1. Vitellogenin

Comparisons among pollen substitutes after one week (W1) showed that bees fed with UBee had *vg* levels significantly higher than those with Apitir (*p* = 0.004) but not significantly different than the bees fed with TAA (Appendix A). Then, while the *vg* levels remained constant in the bees fed with UBee, they increased with Apitir in W2 bees. Lastly, *vg* levels in surviving W3 bees fed with UBee dropped to similar levels as those fed only sugar syrup. As expected, *vg* levels in bees fed only carbohydrates were significantly lower than those of NEBs and bees fed with different pollen substitutes during the first two weeks of chronological age, a period corresponding to the typical nurses’ physiological development on a protein or amino acid-rich diet (Figure 5).

#### 3.3.2. Major Royal Jelly Protein 1

In contrast with the *vg*, the analysis of the *mrjp1* expression pattern shows that bees fed with the free amino acid diet (TAA) had the highest expression level, with significant differences with UBee (*p* = 0.03), but not with Apitir (Appendix A). After two weeks, however, Apitir had the highest *mrjp1* levels, with a significant difference in bees fed with UBee (*p* = 0.025). Like what was observed with *vg*, *mrjp1* levels dropped after three weeks to levels similar to bees fed only sugar (Figure 6).

#### 3.3.3. Deformed Wing Virus

During the first week, the bees that were fed the free amino acid diet (TAA) had the highest levels of DWV, showing significant differences compared to UBee (*p* = 0.006) and Apitir (*p* = 0.002) (Appendix A) (Figure 7). In contrast, DWV levels were low in bees that ingested UBee and intermediate in bees that consumed Apitir. In the second week, there were no significant changes between the two groups of surviving bees fed with pollen substitutes compared to the first week. DWV levels in bees fed UBee were the lowest, with significant differences with Apitir (*p* = 0.034). However, during the third week, DWV levels in bees fed UBee showed a significant increase, reaching levels similar to those observed during the first week with the free amino acid (TAA) diet. On the other hand, it is important to note the DWV levels in bees that were exclusively fed carbohydrates. Bees fed with sugar syrup displayed low DWV levels in the first week, which were significantly lower than those fed with TAA (*p* < 0.001) and Apitir (*p* = 0.003) but were not significantly different from those fed UBee (Appendix A). However, the DWV levels continued to increase as the bees reached 2 and 3 weeks old, although they did not exceed the DWV levels of surviving bees fed UBee at three weeks (W3-Sugar vs. W3-UBee *p* = 0.014). On the other hand, the initial DWV levels in NEBs were lower compared to bees fed TAA (*p* < 0.0001 and Apitir *p* = 0.003) but were similar to those observed in bees fed sugar syrup or Ubee during the first week. Subsequently, DWV levels were higher in the surviving bees fed sugar and pollen substitutes (Appendix A). However, the DWV levels in bees fed UBee during the second week were an exception to this pattern, with lower levels compared to NEBs (*p* = 0.045). 

## 4. Discussion

In this study, we compared the effects of pollen substitutes with varying amino acid availability on honey bee nutrition and the health of caged bees. Specifically, we examined how pollen substitutes made from induced plants (IPs) differed from those composed of free amino acids (FAAs) in their impact on bee survival, hemocyte count, the expression of two nutritional biomarkers, and levels of DWV. We found significant differences in the survival of bees fed different pollen substitutes. Bees fed a diet containing only IPs had higher survival than those fed diets containing FAAs. This result suggests a link between the consumption of FAAs and increased mortality. Our study also revealed that these differences in survival were associated with variations in food intake and water consumption: bees on a diet with IPs ate less food but drank more water than those with a diet containing FAAs. We hypothesize that FAAs have a phagostimulant effect, leading to a higher intake of TAA and Apitir. This proposal is supported by evidence showing that bees can detect changes in the concentration of free amino acids in both liquid [49] and solid solutions [70]. However, it is likely that the increased levels of DWV, rather than the higher food intake, had a more substantial impact on the mortality of the bees fed diets containing FAAs. Further studies are required to uncouple confounding factors in our experimental design, including the effects of the physical state (solid vs. liquid) and amino acid availability (IPs vs. FAAs) of the diet tested.

Hemocytes are essential for the cellular immune response in insects functioning in processes such as phagocytosis, encapsulation, and nodule formation [71]. Prior research has shown that consuming pollen increases hemocyte count [72], whereas feeding with a pollen substitute reduces it [73]. This study explored how diets with differing amino acid availability affect hemocyte counts over time. Our results show a statistically significant decline in hemocyte numbers with age. Newly emerged bees had the highest hemocyte counts, which gradually decreased over subsequent weeks in surviving bees. This age-related decline aligns with previous reports of lower hemocyte counts in foraging bees [74,75] since task allocation in honey bees is age-dependent [76]. On the other hand, bees fed pollen substitutes, particularly those given IPs (Ubee), tended towards higher hemocyte numbers compared with the sugar control group. However, the difference was not statistically significant. This result suggests that the availability of amino acids alone may not be sufficient to modulate hemocyte production, at least under the conditions tested. As previously hypothesized, other nutritional components found in natural pollen—such as phytosterols—could be required to sustain or stimulate hemocyte development [72].

The present study provides new information on the effect of amino acid availability in pollen substitutes and their effects on bee nutrition and health. We did not find significant differences in *vg* levels between bees that consumed a diet of FAAs and those that consumed pollen substitutes containing IPs alone or combined with FAAs. However, *mrjp1* levels were significantly higher in bees that ingested the FAA diet compared to those that consumed diets containing IPs. These findings have practical implications for beekeeping and physiological significance. First, they suggest that the combined use of both biomarkers could help characterize the nutritional value of different pollen substitutes more effectively. Second, our results suggest that the availability of amino acids affects the expression of nutritionally regulated genes differently, with *mrjp1* being more strongly upregulated by amino acid availability than *vg*. The physiological mechanisms behind these differences are currently unknown. However, we hypothesize that the levels of amino acids, which serve as nutritional cues, may prompt a quicker production of royal jelly proteins in the hypopharyngeal glands compared to the synthesis of Vg in the fat bodies. This idea is consistent with the proposed role of Vg in storing nutritional resources in the bodies of sterile workers [77]. This study shows that nutritional biomarkers, such as those used in this study, are valuable for characterizing different pollen substitutes. However, additional methods for assessing dietary utilization, such as analyzing body protein and lipid content, should also be included for a comprehensive evaluation of different diets. 

Our results also show an important effect of amino acid availability on DWV levels. Bees fed diets containing free amino acids had high DWV levels during the first week. During the second week, DWV levels were not measured in bees fed TAA due to their high mortality, but levels with Apitir were still higher than those fed exclusively IPs (UBee). These results suggest a positive relationship between the availability of free amino acids and a fast increase in viral infection. However, DWV levels increased after three weeks in UBee to levels as high as those with TAA in the first week. Thus, although ingesting all pollen substitutes induced high levels of DWV, this increase was delayed by two weeks in the diet consisting exclusively of IPs. Since high DWV titers are linked to bee mortality [78], this delay could be advantageous in maintaining the colony’s population. However, further studies at colony level are required to support this hypothesis.

The comparison between the bees fed with sugar syrup and those fed with pollen substitutes reveals valuable insights into the relations between bee nutrition and physiological development. Firstly, consistent with their utilization as nutrition biomarkers, the expression levels of *vg* and *mrjp1* were low in the control group fed only carbohydrates. Secondly, *vg* and *mrjp1* are also markers of physiological development, which is closely linked to behavioral development [51]. Low levels of these genes in carbohydrate-fed young bees suggest they experienced an early transition to the physiological foraging state due to acute nutritional stress. In contrast, bees fed with pollen substitutes showed high levels of these genes during the first two weeks, indicating a nurse-like physiological state. Thirdly, DWV levels were low in sugar-fed bees in the first week but increased after two weeks. This pattern partially resembles that reported by DeGrandi-Hoffman et al. (2010), where DWV levels increased with age in sugar-fed bees, although these levels were still lower compared with bees fed pollen or a plant-based pollen substitute in 11-day-old bees [60]. Finally, both the quick age-related increase in DWV levels in bees fed with carbohydrates and the delayed increase in viral levels observed in 3-week-old bees fed IP pollen substitutes could be influenced by immunological changes associated with the nurse–forager physiological transition, given that foraging is associated with increased DWV levels at the colony level [51]. Further studies are needed to understand the effects of amino acid availability and nutritionally induced changes in bees’ physiology and immunity.

We are unaware of previous studies reporting increased viral levels after ingesting pollen substitutes. However, other researchers found increased DWV levels associated with pollen ingestion in cages [55] and colony-level studies [51]. Additionally, Branchiccela et al. (2019) found that bees fed with a high-nutritional-value polyfloral pollen had higher DWV levels than those fed with a low-nutritional-value monofloral pollen [61]. The diverse results from studies examining the impact of nutrition on DWV levels suggest the existence of previously overlooked factors. We propose that the effect of nutrition on viral infection may vary depending on the initial levels of DWV in the bees being studied. According to this idea, there would be a threshold of viral infection where moderate and high levels of DWV would lead to increased viral replication if the infected bees ingest food rich in proteins or amino acids. However, the levels of virus in the experimental bees may not be determined solely by the infestation levels of *Varroa* in the colonies at the time of their use in nutritional treatments, but also by past *Varroa* infestation levels. Evidence shows that DWV titers in the colony decrease significantly after the acaricide treatment, although viral levels gradually increase without *Varroa* [79]. In our experiments, the donor colonies had low *Varroa* infestation levels (<1%) at the beginning of the experiment. However, these colonies had high *Varroa* levels (>5%) before being treated with a thymol-based acaricide a month before the experiment. Considering that cleaning DWV infection in the colony involves population turnover and takes approximately six weeks [79], it is likely that the bees used in our experiment still had moderate DWV levels. We used in our study a relative quantification method, which prevented us from determining the copy number of DWV in the analyzed samples and its clinical significance. If our hypothesis is correct, the initial levels of DWV in the NEBs used in our study, prior to the application of treatments, may have exceeded the threshold necessary to trigger viral replication when fed a diet rich in protein or amino acids.

## 5. Conclusions

This study compared how pollen substitutes made from IPs differed from those composed of mixed free amino acids in terms of their impact on the expression of two genes encoding important nutritionally regulated proteins—*vg* and *mrjp1*—as well as a prevalent bee virus (DWV). Regardless of their differences in amino acid availability, we found no significant variations in *vg* expression in bees fed the different pollen substitutes. However, we did observe significant differences in *mrjp1* expression between the diets made from FAAs and those containing IPs. These findings suggest that utilizing a combination of nutritional biomarkers could be beneficial in assessing the nutritional value of different types of pollen substitutes. Similarly to previous reports on pollen ingestion, our findings indicate that consuming pollen substitutes can also lead to elevated DWV levels. However, there were remarkable differences in the timing of when the bees fed the different pollen substitutes exhibited high DWV levels: those consuming a diet of FAAs showed increased DWV levels after one week, while bees fed a diet of IPs exhibited elevated DWV levels after three weeks. Possible explanations for these results include differences among diets in terms of the amino acids’ intestinal absorption and nutritionally induced changes in honey bee physiological development. The impact of nutrition on viral infections in bees may differ based on the initial levels of DWV present in the bees being studied.

## Figures and Tables

**Figure 1 insects-16-00375-f001:**
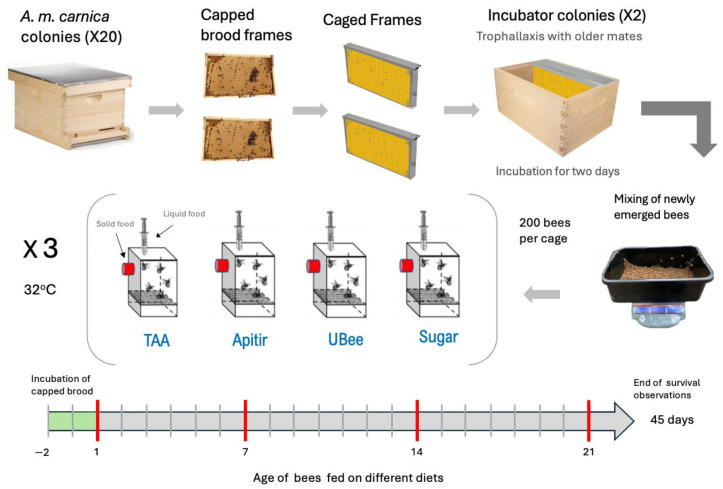
Timeline of experimental design. The arrows’ red lines indicate the ages at which the bees were collected for molecular analysis (n = 9). Bee survival and food consumption were recorded for 45 days.

**Figure 2 insects-16-00375-f002:**
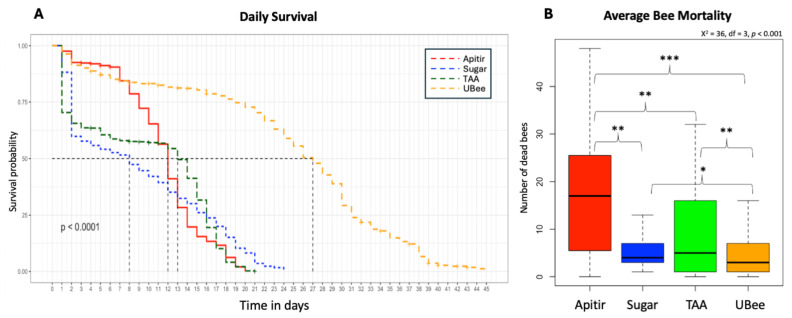
Survival probability and average mortality of honey bees on different types of pollen substitutes (**A**). Kaplan–Meier survival probability model. The *y*-axis represents the survival probability, while the *x*-axis shows the age of the bees. The dotted black lines indicate the day when 50% of the bees died in each treatment (LD50) (**B**). Average mortality of bees given different treatments. Kruskal–Wallis test levels of significance: *p* < 0.05 (*), *p* < 0.01 (**), and *p* < 0.001 (***).

**Figure 3 insects-16-00375-f003:**
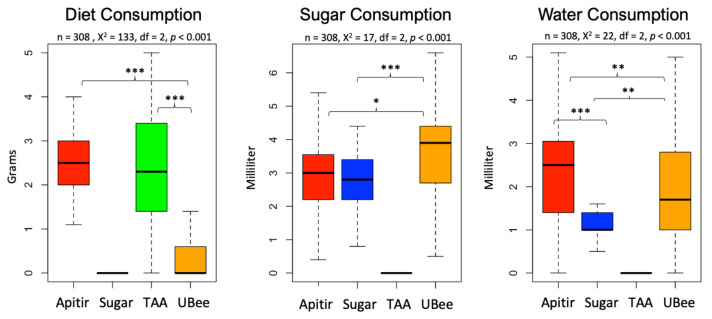
Honey bee diet consumption. The y-axes indicate diet consumption in grams. Box plots represent the first and third quartiles ranges, with a line representing the median. Whiskers include the values of 90% of the samples. Level of significance: *p* < 0.05 *, *p* < 0.01 **, and *p* < 0.001 *** (Kruskal–Wallis test).

**Figure 4 insects-16-00375-f004:**
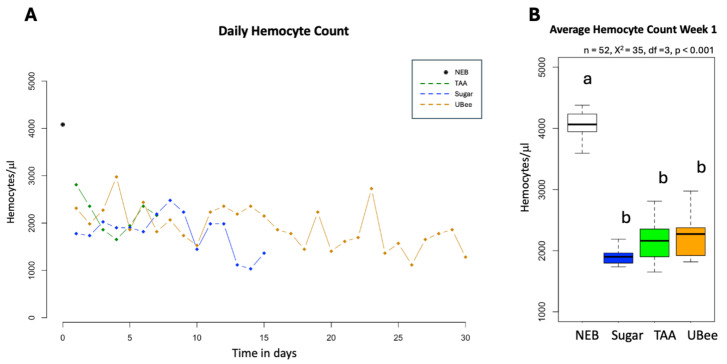
Hemocyte count. (**A**) Daily hemocyte count. The *y*-axis represents the hemocyte count per μL of hemolymph, while the *x*-axis shows the time in days (*n* = 82). (**B**) Average hemocyte count in 1–7-day-old bees. The *x*-axis represents the groups of bees given different treatments. Hemocyte counts were not conducted for the bees fed with Apitir due to a limited number of surviving individuals. Different letters (a,b) indicate significant differences (Krustal–Wallis test).

**Figure 5 insects-16-00375-f005:**
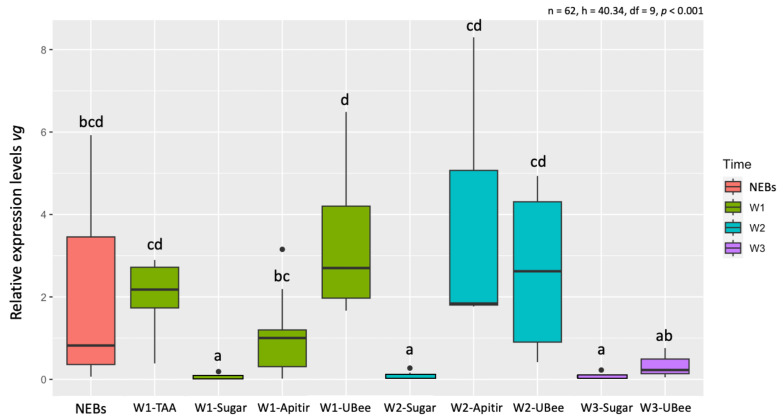
Relative *vg* mRNA levels. The *y*-axis represents log10-transformed relative expression levels, while the *x*-axis displays the groups of bees fed with different pollen substitutes collected after 1, 2, and 3 weeks (W1-W3). Newly emerged bees (NEBs) were used as a control. The ribosomal protein S5 (*rps5*) gene was used as an internal control for qPCR measurements. The graphs’ boxes show the first and third quartile ranges, with a line representing the median. The whiskers encompass the values of 90% of the samples. Black dots represent single outliers. Expression analysis was not conducted on the TAA diet during W2 and W3 because of a limited number of surviving individuals. Lowercase letters indicate significant differences between treatments (*p* < 0.05).

**Figure 6 insects-16-00375-f006:**
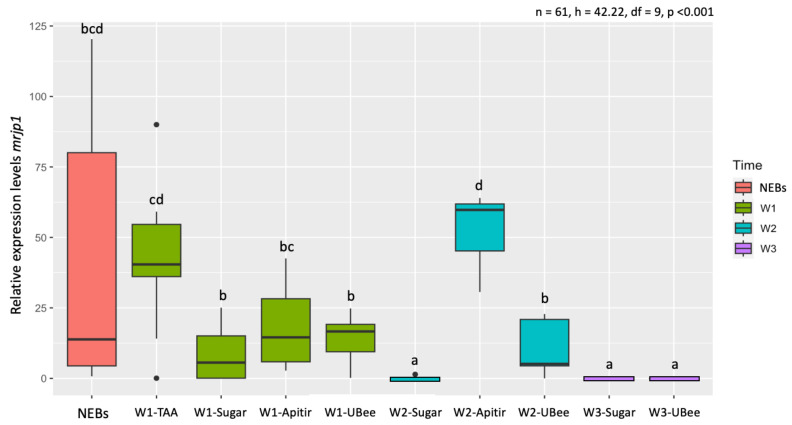
Relative *mrjp1* mRNA levels. The *y*-axis represents log10-transformed relative expression levels. The *x*-axis displays the groups of bees fed with different pollen substitutes. Black dots represent single outliers. Expression analysis was not conducted on the TAA diet during W2 and W3 because of a limited number of surviving individuals. Lowercase letters indicate significant differences between treatments (*p* < 0.05).

**Figure 7 insects-16-00375-f007:**
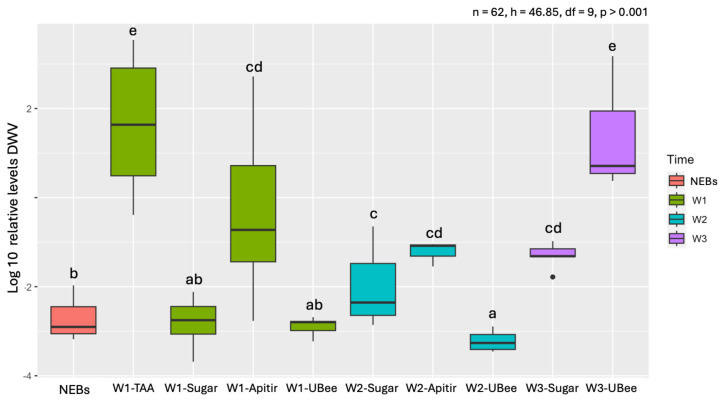
Relative DWV mRNA levels. The *y*-axis represents log10-transformed relative expression levels. The *x*-axis displays the groups of bees fed with different pollen substitutes. Black dots represent single outliers. Expression analysis was not conducted on the TAA diet during W2 and W3 because of a limited number of surviving individuals. Lowercase letters indicate significant differences between treatments (*p* < 0.05).

## Data Availability

The original contributions presented in this study are included in the article/Appendix A. Further inquiries can be directed to the corresponding author.

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
