# Peer review of "The Effects of Artificial Diets Containing Free Amino Acids Versus Intact Proteins on Biomarkers of Nutrition and Deformed Wing Virus Levels in the Honey Bee"

_insects, 2025, doi:10.3390/insects16040375_

Round 1
Reviewer 1 Report
Comments and Suggestions for Authors
The presented study deals with the comparison of artificial diets for honey bees and their impact on biomarkers of nutrition. The main goal is to compare the effect of a free amino acid diet and a diet consisting of intact proteins. The study follows trends in the honey bee research and protein supplementation in beekeeping. Although I like the idea of using a mixture of individual essential amino acids as easy and affordable feed for bees, I have a few major concerns about the study, especially concerning the experimental part.
Firstly, I do not consider the chosen measured parameters sufficient for supporting the author’s theory. The authors decided to measure the gene expression of only two nutritionally regulated genes mrjp1 (major royal jelly protein 1) and vg (vitellogenin), as main nutrition biomarkers. I agree with the importance of those two proteins in the nutritional status of bees; however, I would strongly suggest to support these parameters and the author’s theory with other genes connected to bee nutrition and digestibility of selected diets - for example, storage proteins and essential sources of amino acids hexamerins (Martins et al., 2008 https://doi.org/10.1016/j.jinsphys.2008.03.009) or antioxidant enzymes – superoxide dismutase (SOD), thioredoxin reductase (TrxR) (Frunze et al., 2024, https://doi.org/10.3390/ijms25084271 ).
Second, for studies concerning bee nutrition, it is important to measure basic physiological parameters such as total protein, lipid, and carbohydrate content to reveal the effect of food intake on metabolism – in this case, especially on total protein content to see the effect of amino acids utilization into proteins. Some previous studies also showed increased mortality of bees fed only with protein diet or essential amino acids solution (EAAs), suggesting not efficient conversion EAAs into energy (Pirk et al. 2010, https://doi.org/10.1051/apido/2009055 ; Dussutour and Simpson 2012, DOI: 10.1098/rspb.2012.0051, Paoli et al., 2014, https://doi.org/10.1007/s00726-014-1706-2). Therefore, these parameters should be especially followed when testing free amino acid diets.
I also have one major comment on the section focused on the hemocytes. I am missing the link between the total hemocyte count and the rest of the article. If this parameter is concluded within the experimental part of the work, there has to be a link to the main story of the article and to discussion. Hemocytes are an important part of honey bee immunity, therefore, I would suggest adding some connection of hemoctes to the immunity (and DWV levels) and nutrition into the article. If there is a measured parameter such as hemocyte count, I would also suggest connecting it to gene expression levels of antimicrobial molecules, e. g., antimicrobial peptides in the experimental part.
I added also some other comments and questions, which I have described specifically below:
Introduction
Line 94: I would suggest to cite all mentioned studies on FAAs, EAA and IPs ingestion after the first sentence.
Line 100-102: The ability of bees to metabolize a high protein diet or free amino acids solutions (FAAs) should be mentioned in this section.
Materials and methods
Line 146: I would suggest using a different word than homogenized (mixed, pooled…). Homogenization indicates mechanically damaging the bees.
Line 151: There should be added also some basic characterization of other diets (total proteins, lipids etc.) if possible.
Results
Within the result section, I missed the number of replicates for each experiment (and individual groups) in the graphs.
Line 281: The comparison of gene expression levels is a little bit confusing. There are some groups missing in W2 and W3 – I suppose that it is due to mortality and the number of available bees. If this is the case, would not be better only to compare W1 graphically where data for all groups are available? This fact should be also mentioned in the description of the graph for easier understanding.
Discussion
Line 343: I would be careful with recommending of FAAs to practical implications for beekeeping due to the almost two times faster mortality of bees in TAA and Apitir group compared to UBee.
Line 365: The fact that bees in TAA and Apitir group had quite high mortality should be discussed more in a biochemical context. As it was mentioned in previous studies – there are concerns about efficient amino acids utilization when the diet is very high in proteins or amino acids. That is also the reason why it would be really helpful to have data about protein, lipids, and carbohydrate content in bees and even in different of diets. Also, information about the defecation of bees in the cages would provide some hints about EAA conversion in bee´s organism.
Line 383: “Low levels of these genes in carbohydrate fed young bees suggest they have experienced an early transition to the physiological foraging state due to acute nutritional stress.” For gene expression of storage proteins, it is necessary to have protein intake in diet. Low levels of those genes do not need to necessarily indicate early transition. This result could be also due to malnutrition and insufficient sources for protein synthesis (for example study of Paiva et al., 2019 https://doi.org/10.1007/s13592-019-00661-4,)
Line 398: I really miss the connection between DWV and some immune parameters. Even hemocyte count should be discussed here. Activation of immune system is very influenced by level of DWV and should not be overlooked in the context of DWV levels, energy demand on immune activation and the availability of protein in diet.
Author Response
The presented study deals with the comparison of artificial diets for honey bees and their impact on biomarkers of nutrition. The main goal is to compare the effect of a free amino acid diet and a diet consisting of intact proteins. The study follows trends in the honey bee research and protein supplementation in beekeeping. Although I like the idea of using a mixture of individual essential amino acids as easy and affordable feed for bees, I have a few major concerns about the study, especially concerning the experimental part.
Firstly, I do not consider the chosen measured parameters sufficient for supporting the author’s theory. The authors decided to measure the gene expression of only two nutritionally regulated genes mrjp1 (major royal jelly protein 1) and vg (vitellogenin), as main nutrition biomarkers. I agree with the importance of those two proteins in the nutritional status of bees; however, I would strongly suggest to support these parameters and the author’s theory with other genes connected to bee nutrition and digestibility of selected diets - for example, storage proteins and essential sources of amino acids hexamerins (Martins et al., 2008 https://doi.org/10.1016/j.jinsphys.2008.03.009) or antioxidant enzymes – superoxide dismutase (SOD), thioredoxin reductase (TrxR) (Frunze et al., 2024, https://doi.org/10.3390/ijms25084271 ).
R: We thank the reviewer for this important comment. The rationale for using these two nutritional biomarkers is grounded on previous studies where we analyzed the effect of an acute nutritional state, such as pollen restriction (Corona et al., 2023) on gene expression. In that study, we analyzed the expression of several potential nutritional markers, including key elements of the insulin signaling pathway. The only genes that showed significant nutritional regulation in both young nurses and foragers were vitellogenin and mrjp1. These findings emphasize that only a few nutritionally regulated genes are consistently affected across different behavioral stages and chronological ages. While other genes, such as hex70a, could also demonstrate robust nutritional regulation, as suggested by the reviewer, the aim of the present study is not to conduct an exhaustive search for nutritionally regulated genes. Instead, we focus on the markers previously shown to be consistently regulated by various diets in most conditions. In fact, unlike many similar articles that analyze only vitellogenin as a nutritional biomarker, this study proposes to include a second robust biomarker, mrjp1, to enhance the practical nutritional characterization of the different diets used in the beekeeping industry.
Second, for studies concerning bee nutrition, it is important to measure basic physiological parameters such as total protein, lipid, and carbohydrate content to reveal the effect of food intake on metabolism – in this case, especially on total protein content to see the effect of amino acids utilization into proteins. Some previous studies also showed increased mortality of bees fed only with protein diet or essential amino acids solution (EAAs), suggesting not efficient conversion EAAs into energy (Pirk et al. 2010, https://doi.org/10.1051/apido/2009055 ; Dussutour and Simpson 2012, DOI: 10.1098/rspb.2012.0051, Paoli et al., 2014, https://doi.org/10.1007/s00726-014-1706-2). Therefore, these parameters should be especially followed when testing free amino acid diets.
R: Our understanding from these reports is that high concentrations of proteins or amino acids can be toxic to bees, rather than indicating that amino acids are not efficiently converted into energy or incorporated into proteins. (please note that in our experiment we don't use EAA but total amino acids). Our objective was not to report on the diet optimization process in this study. However, during our diet optimization experiments, we also found that high amino acids concentrations are toxic to bees. The optimal total amino acid concentration in our diet was 22 mM, which aligns with the ranges observed in other studies on insect diets (https://doi.org/10.1038/nature08619). Additionally, the amino acid to carbohydrate ratio in our TAA diet is 1:75, which falls within the optimal ratios found by Paoli et al. in 2014. In conclusion, the amino acid concentration used in our TAA diet is unlikely associated to poor amino acid conversion into proteins. However, we acknowledge that conducting a protein analysis of the tested bees would have been appropriate. A sentence was added about it in the discussion (see lines 107-113).
“In honey bees, diets with high concentrations of IPs or EAAs have been associated with increased mortality, suggesting that the bees may be unable to metabolize these components at high levels[51, 52]. An intact protein-to-carbohydrate (IP:C) ratio of 1:3 produced the highest ovarian activation in bees that were fed royal jelly, although higher IP:C ratios were associated with increased mortality [52]. On the other hand, the optimal EAA to carbohydrate (EAA:C) ratio varies with the bee's age, ranging from 1:50 in nurse bees to 1:75 in foragers. However, higher EAA:C ratios also increased mortality [52]. These studies reveal that, contrary to common assumptions, pollen substitutes with high concentrations of proteins or amino acids may create a nutritional imbalance that is harmful to bee health”.
I also have one major comment on the section focused on the hemocytes. I am missing the link between the total hemocyte count and the rest of the article. If this parameter is concluded within the experimental part of the work, there has to be a link to the main story of the article and to discussion. Hemocytes are an important part of honey bee immunity, therefore, I would suggest adding some connection of hemoctes to the immunity (and DWV levels) and nutrition into the article. If there is a measured parameter such as hemocyte count, I would also suggest connecting it to gene expression levels of antimicrobial molecules, e. g., antimicrobial peptides in the experimental part.
R: We addressed this concern below on the discussion part.
I added also some other comments and questions, which I have described specifically below:
Introduction
Line 94: I would suggest to cite all mentioned studies on FAAs, EAA and IPs ingestion after the first sentence.
R: Corrected as suggested. We added all the references mentioning studies on FAAs and IPs in mammals as suggested (line 99).
Line 100-102: The ability of bees to metabolize a high protein diet or free amino acids solutions (FAAs) should be mentioned in this section.
R: We appreciate the reviewer's valuable feedback. In response, we have added a new section summarizing the limitations of bees in metabolizing high concentrations of proteins and amino acids. Additionally, we have emphasized the significance of the ratios of proteins and amino acids to carbohydrates in honey bee nutrition. (lines 101-110).
Quote:
“In honey bees, diets with high concentrations of IPs or EAAs have been associated with increased mortality, suggesting that the bees may be unable to metabolize these components at high levels[50, 51]. An intact protein-to-carbohydrate (IP:C) ratio of 1:3 was shown to produce the highest ovarian activation in bees that were fed royal jelly, although higher IP:C ratios were associated with increased mortality [51]. On the other hand, the optimal EAA to carbohydrate (EAA:C) ratio varies with the bee's age, ranging from 1:50 in nurse bees to 1:75 in foragers. However, higher EAA:C ratios were also associated with increased mortality [51]. These studies unveil that, contrary to common assumptions, pollen substitutes containing high concentrations of proteins or amino acids could lead to a nutritional imbalance detrimental to bee health”.
Materials and methods
Line 146: I would suggest using a different word than homogenized (mixed, pooled…). Homogenization indicates mechanically damaging the bees.
R: We have changed the word "homogenized" to "mixed" (line 157) and Figure 1.
Line 151: There should be added also some basic characterization of other diets (total proteins, lipids etc.) if possible.
R: Now we included the basic composition of the three diets in Table S1.
Results
Within the result section, I missed the number of replicates for each experiment (and individual groups) in the graphs.
R: We had included the “n” number on the top of graphics.
Line 281: The comparison of gene expression levels is a little bit confusing. There are some groups missing in W2 and W3 – I suppose that it is due to mortality and the number of available bees.
R: Yes, indeed this is the case. We clarified this, by adding “Expression analysis was not conducted on the TAA diet during W2 and W3 because of a limited number of surviving individuals” in the figure legends.
If this is the case, would not be better only to compare W1 graphically where data for all groups are available? This fact should be also mentioned in the description of the graph for easier understanding.
R: We appreciate this feedback. Indeed, we also considered only showing the graph with the results for one week across all groups. However, the story we want to tell would be incomplete if we did not highlight the increased DWN in week 3 with the intact protein diet (Ubee). We believe it important to demonstrate that this virus increase also occurs even with the diet initially appearing ideal during the first week of the experiment.
Discussion
Line 343: I would be careful with recommending of FAAs to practical implications for beekeeping due to the almost two times faster mortality of bees in TAA and Apitir group compared to UBee.
R: Please note that in this part, we refer to the simultaneous use of vg and mrjp1 as nutritional biomarkers for assessing the quality of pollen substitutes. We do not intend to endorse the use of TAA, even though this diet has been applied successfully in various conditions (10.1016/j.heliyon.2025.e42042; 10.1016/j.jip.2022.107830). We expect the TAA diet to be utilized only when DWV falls below a certain threshold level, as discussed below.
Line 365: The fact that bees in TAA and Apitir group had quite high mortality should be discussed more in a biochemical context. As it was mentioned in previous studies – there are concerns about efficient amino acids utilization when the diet is very high in proteins or amino acids.
R: First, as highlighted in the articles recommended by the reviewer, it is evident that high concentrations of amino acids or proteins can lead to nutritional imbalances. We also observed this phenomenon in our experience. However, it is important to note that both diets containing FAAs (Apitir and TAA) were already experimentally optimized. Specifically, regarding TAA, its amino acid to carbohydrate ratio (1:75) should not be linked to a nutritional imbalance (Paoli et al., 2014). Second, a nutritional unbalance is associated with mortality, but so far, we know that it has not been associated with increased DWV levels. In any case, the effect of nutritional imbalance on DWV levels should be further studied, but we consider that the effect observed in this study is not due to this cause.
That is also the reason why it would be really helpful to have data about protein, lipids, and carbohydrate content in bees and even in different of diets. Also, information about the defecation of bees in the cages would provide some hints about EAA conversion in bee´s organism.
R: We agree that could have been helpful about the composition of the bees on different time and with different diets. Regrettably, our experimental design did not consider such biochemical tests. We included a sentence in the discussion, at the end of the paragraph discussing the use of molecular markers of nutrition, to acknowledge this limitation of our study (lines 411-415)
Quote:
“This study shows that nutritional biomarkers, like those used in this study, are valuable for characterizing different pollen substitutes. However, additional methods for assessing dietary utilization, such as analyzing body protein and lipid content, should also be included for a comprehensive evaluation of different diets”.
Line 383: “Low levels of these genes in carbohydrate fed young bees suggest they have experienced an early transition to the physiological foraging state due to acute nutritional stress.” For gene expression of storage proteins, it is necessary to have protein intake in diet. Low levels of those genes do not need to necessarily indicate early transition. This result could be also due to malnutrition and insufficient sources for protein synthesis (for example study of Paiva et al., 2019 https://doi.org/10.1007/s13592-019-00661-4,).
R: This is an important topic that has not been sufficiently addressed in the literature. The beauty of using vg and mrjp1 is that, in addition to being biomarkers of nutrition, they are even better behavior markers. This characteristic implies that the bee will be a physiological forager as long as these genes are expressed at low levels. (10.1073/pnas.0701909104, 10.1073/pnas.0800630105, 10.1073/pnas.0701909104).
Line 398: I really miss the connection between DWV and some immune parameters. Activation of immune system is very influenced by level of DWV and should not be overlooked in the context of DWV levels, energy demand on immune activation and the availability of protein in diet.
R: We don't know if there is a connection between hemocytes and DWV, but this topic is under current investigation.
Even hemocyte count should be discussed here.
R: Given the lack of notable differences among the pollen substitutes, we considered it might be unnecessary to include hemocyte discussion in our paper. Nonetheless, the reviewer's feedback has prompted us to add a brief discussion on this topic (380-394).
Hemocytes are essential for the cellular immune response in insects functioning in processes such as phagocytosis, encapsulation, and nodule formation [71]. Prior research has shown that consuming pollen increases hemocyte count [72], whereas feeding with a pollen substitute reduces it [73]. This study explored how diets with differing amino acid availability affect hemocyte counts over time. Our results show a statistically significant decline in hemocyte numbers with age. Newly emerged bees had the highest hemocyte counts, which gradually decreased over subsequent weeks in surviving bees. This age-related decline aligns with previous reports of lower hemocyte counts in foraging bees [74, 75], since task allocation in honey bees is age-dependent [76]. On the other hand, bees fed pollen substitutes, particularly those given IPs (Ubee), tended towards higher hemocyte numbers compared with the sugar control group. However, the difference was not statistically significant. This result suggests that the availability of amino acids alone may not be sufficient to modulate hemocyte production, at least under the conditions tested. As previously hypothesized, other nutritional components found in natural pollen—such as phytosterols—could be required to sustain or stimulate hemocyte development [72].
Please note that while we discussed the results of the hemocytes, we realized it is more appropriate to present data up to the first week of the experiment rather than aggregating it over the entire four-week period, especially since some groups of bees did not survive. We observed a time-dependent reduction in hemocyte counts, mainly affecting the longest-lived group (Ubee). By aggregating all the data over time, as we initially did, we inadvertently lowered the Ubee levels. However, this new approach to analyzing hemocytes does not change our previous conclusions, as there are still no significant differences among the diets. To capture the results better, we have now included two graphs. The first one (Panel A) plots the effects of the diets on daily hemocyte count during a month. The second one shows the average hemocyte count during the first week when it is possible to compare all the analyzed groups.
Reviewer 2 Report
Comments and Suggestions for Authors
This is an interesting paper, within the scope of Insects. It is mostly well compiled and well presented. Some details need to be corrected and clarified.
I attach the document, annotated with comments for the attention of the authors. Main comments are given below.
In the methods, L154-158, the numbering of the treatment groups is unclear from the way it is presented, but also this numbering does not seems to be used anywhere else in the article.
L151 and L169: what do you mean by “Age collections”? Can you make this clearer?
L183-184: the wording needs revisited. What did you find the difference of?
L235: I think you have not yet defined W1. You do define it later on.
L241: “Non-parametric data” would better as “Non-normal data”.
Figure 2 legend and plot B: you say this plots average bee mortality, but it is unclear where averages were used. What is the underlying data? You also do not comment in the text on what plot B tells us.
Figure 3 legend: you refer to the “first and third interquartile ranges” but mean the “first and third quartiles”. Also how do the whiskers include 90% of the values? Do you mean from the start of the lower whisker to the top of the upper whisker, and did you exclude outliers? The legend should also refer to the Kruskal-Wallis test.
L273: the > sign should be a < sign.
Figure 4 legend: you say this plots average bee mortality, but it is unclear where averages were used. What is the underlying data? It would be clearer to state this in the legend, possibly with sample sizes as well.
L285: you refer to Table S3. There are many test results given in this table. Did you allow for multiple testing in Table S3? If so, you should say so in the legend of the table.
Figure 5 legend: L295: you should refer to NEBs at week 0 (W0) as this notation is used in the plot. Also see comments about quartiles and whiskers for Figure 3 legend – the same applies here.
L324: you refer to Table S4, but there is no Table S4. You mean Table S3.
L441-442: you have not put in the details of the supplementary materials.
References- in a few places there are capitals where there should not be and in another place lower case where there should not be – see the annotated paper.
These points and some minor points of wording are also marked on the document.

Author Response
This is an interesting paper, within the scope of Insects. It is mostly well compiled and well presented. Some details need to be corrected and clarified.
R: Thank you for your positive feedback.
I attach the document, annotated with comments for the attention of the authors. Main comments are given below.
R: We greatly appreciate this comments that have improved our manuscript
In the methods, L154-158, the numbering of the treatment groups is unclear from the way it is presented, but also this numbering does not seems to be used anywhere else in the article.
- The abbreviations for the treatments (T1-T4) were removed as it were unnecessary and confusing.
L151 and L169: what do you mean by “Age collections”? Can you make this clearer?
R: Age collections” refers to collections conducted with bees of varying ages. We welcome suggestions for a more suitable term.
L183-184: the wording needs revisited. What did you find the difference of?
R: We appreciate you bringing this issue to our attention. This section has been revised.
L235: I think you have not yet defined W1. You do define it later on.
We defined W1 before on the 2.2. section “honey bees and age collections” Lines XX=XX.
L241: “Non-parametric data” would better as “Non-normal data”.
R: Corrected as advised.
Figure 2 legend and plot B: you say this plots average bee mortality, but it is unclear where averages were used. What is the underlying data? It would be clearer to state this in the legend, possibly with sample sizes as well. You also do not comment in the text on what plot B tells us.
R: Each box plot represents the overall average of counted dead bees throughout the experiment. The median is also given for each treatment. We have also included the number of observations (n) at the top of the graph.
Figure 3 legend: you refer to the “first and third interquartile ranges” but mean the “first and third quartiles”. Also how do the whiskers include 90% of the values? Do you mean from the start of the lower whisker to the top of the upper whisker, and did you exclude outliers? The legend should also refer to the Kruskal-Wallis test.
R: We corrected to “quartiles”. We don't exclude outliers, as none of the values exceeded two standard deviations (SD) from the mean. In contrast, we removed outlier exceeding 2 SD from the mean in the gene expression analysis as indicated in materials ad method section (lines XX-XX). We also clarified that the statistical analysis was conducted using the Kruskal-Wallis test.
L273: the > sign should be a < sign.
R: Corrected. Thank you.
L285: you refer to Table S3. There are many test results given in this table. Did you allow for multiple testing in Table S3? If so, you should say so in the legend of the table.
R: This table shows the paired comparisons among groups, a analyses conducted with the Mann-Whitney test. This is indicated in the legend.
Figure 5 legend: L295: you should refer to NEBs at week 0 (W0) as this notation is used in the plot. Also see comments about quartiles and whiskers for Figure 3 legend – the same applies here.
R: We corrected as indicated in the Figure 5-7.
L324: you refer to Table S4, but there is no Table S4. You mean Table S3.
R: Corrected.
L441-442: you have not put in the details of the supplementary materials.
R: We don't know how to get this link before submitting the revised manuscript. Is this possible?
References- in a few places there are capitals where there should not be and in another place lower case where there should not be – see the annotated paper.
R: Thank you. This is corrected.
These points and some minor points of wording are also marked on the document.
R: We really appreciate your help.
Reviewer 3 Report
Comments and Suggestions for Authors
The article titled “The effects of artificial diets containing free amino acids versus intact proteins on biomarkers of nutrition and deformed wing virus levels in the honey bee” presents an interesting study on the effects of various pollen substitutes, those with intact protein profiles versus those with free amino acids, on honeybee survival, hemocytes, nutritional status markers, and the presence of deformed wing virus (DWV) as an indicator of bee health.
However, there are several concerns with this article. Overall, the presentation of treatments could be significantly improved. The order in which the results are presented is not consistently followed in the discussion or the conclusion, resulting in confusion.
A major issue is the selection of treatments. It is unclear how the mixtures were chosen to evaluate the hypothesis, specifically, what exactly was intended to be compared: the composition, the availability of amino acids, or the state of the food. Additionally, the introduction does not clearly explain the difference between a diet classified as total amino acids (TAA) and one classified as essential amino acids (EAA). I recommend expanding on this classification for clarity.
Specific Comments
The order of the treatments in the methodology section (Lines 154–159) is somewhat confusing, and the descriptions lack clarity.
Treatment 1: Described as a liquid amino acid mixture; however, the brand of the product is not specified.
Treatment 2: Described as a solid form of soy-based intact proteins (Ultra Bee®).
Treatment 3: Described as a solid form of yeast-based intact proteins supplemented with TAA (Apitir®).
Treatment 4: Consists of a 50% sugar syrup.
Please clarify the information provided between Lines 158 and 159, specifically whether that description relates to Treatment 1. Additionally, the use of the labels T1, T2, T3, and T4 is confusing since these terms are not used elsewhere in the article.
The order and nomenclature are not consistent with what is presented in Figure 1, nor with the order of the graphs shown in the other figures. Moreover, it appears that the “Sugar” treatment is a control; therefore, placing it at the beginning might help readers understand the results better.
There seems to be a lack of analysis of free amino acids across all treatments, which would justify the discussion and analysis of the results. It is not clear which treatment(s) are being designated as containing free amino acids since the methodology and results sections only refer to the treatments as TAA.
Lines 273–275: Although the text mentions a slight increase in hemocyte concentration in bees treated with TAA, the increase is so minimal that it might not be worth discussing.
Figure 3: The consumption of syrup, diet, and water is reported in grams, but the time unit is not specified.
Figure 4: The x-axis is labeled “Hemocitos” in Spanish instead of being translated into English.
The sentence “These findings have practical implications for beekeeping and physiological significance. First, they suggest that the combined use of both biomarkers could help characterize the nutritional value of different pollen substitutes more effectively” is redundant, as it is repeated in lines 343–345.
The discussion starts by addressing the expression of mrjp1 and vg mRNAs, which does not follow the same order as the results. The first results presented pertain to survival percentage and food consumption, and this inconsistency can confuse readers.
Author Response
The article titled “The effects of artificial diets containing free amino acids versus intact proteins on biomarkers of nutrition and deformed wing virus levels in the honey bee” presents an interesting study on the effects of various pollen substitutes, those with intact protein profiles versus those with free amino acids, on honeybee survival, hemocytes, nutritional status markers, and the presence of deformed wing virus (DWV) as an indicator of bee health.
However, there are several concerns with this article. Overall, the presentation of treatments could be significantly improved. The order in which the results are presented is not consistently followed in the discussion or the conclusion, resulting in confusion.
A: Thank you for this useful comment. We have rearranged the order of the topics covered in the discussion to follow the order of the results better.
A major issue is the selection of treatments. It is unclear how the mixtures were chosen to evaluate the hypothesis, specifically, what exactly was intended to be compared: the composition, the availability of amino acids, or the state of the food.
R: The goal of this study was to compare different pollen substitutes with varying amino acid availability. This study did not compare other variables like the physical state of the diets. We acknowledged that these variables should be uncoupled in future studies (lines 376-378).
Quote:
“Further studies are required to uncouple confounding factors in our experimental design, including the effects of the physical state (solid vs. liquid) and amino acid availability (IPs vs. FAAs) of the diet tested”
Additionally, the introduction does not clearly explain the difference between a diet classified as total amino acids (TAA) and one classified as essential amino acids (EAA). I recommend expanding on this classification for clarity.
R: We have revised the introduction to clarify the differences between intact proteins (IP), free amino acids (FAAs), total amino acids (TAA), and essential amino acids (EAA). We have added two sentences:
Lines (88-90)
“These artificial diets typically contain amino acids in two forms: intact proteins (IPs), where amino acids are covalently linked together, and free amino acids (FAAs), in which the amino acids remain separate”.
Quote, Lines (94-96)
“A TAA diet includes all twenty amino acids found in proteins, whereas an EAA diet consists of the ten essential amino acids the body cannot synthesize”.
Specific Comments
The order of the treatments in the methodology section (Lines 154–159) is somewhat confusing, and the descriptions lack clarity.
R: We believe this section will be clearer after the changes we made in the introduction. Nevertheless, we have made some modifications to enhance clarity.
Treatment 1: Described as a liquid amino acid mixture; however, the brand of the product is not specified.
Treatment 2: Described as a solid form of soy-based intact proteins (Ultra Bee®).
Treatment 3: Described as a solid form of yeast-based intact proteins supplemented with TAA (Apitir®).
Treatment 4: Consists of a 50% sugar syrup.
Please clarify the information provided between Lines 158 and 159, specifically whether that description relates to Treatment 1.
Treatment 1 (TAA) is not patented and does not have a commercial name. For simplicity, we have given it a generic name, an abbreviation for Total Amino Acids. We will now provide more details about this diet:
Quote. Lines 168-172.
“Apitir and UBee are proprietary diets available for commercial use. TAA is a diet developed at the USDA Bee Research Laboratory in Beltsville, Maryland. This diet has been previously used in experiments to evaluate the effect of phytochemicals on bee pathogens and stimulate egg laying in honey bee queens confined to small laboratory colonies”.
Additionally, the use of the labels T1, T2, T3, and T4 is confusing since these terms are not used elsewhere in the article. The order and nomenclature are not consistent with what is presented in Figure 1, nor with the order of the graphs shown in the other figures. Moreover, it appears that the “Sugar” treatment is a control; therefore, placing it at the beginning might help readers understand the results better.
R: Thank you for pointing out these issues. We agree that the abbreviations T1-T4 are unnecessary and confusing, and we have removed them. We have also made the diet nomenclature consistent throughout the text and figures. Finally, we clarified in the method section that the sugar-fed group is a control in this experiment.
There seems to be a lack of analysis of free amino acids across all treatments, which would justify the discussion and analysis of the results. It is not clear which treatment(s) are being designated as containing free amino acids since the methodology and results sections only refer to the treatments as TAA.
R: We believe that the modifications made to the introduction and methods sections will enhance the clarity of the analysis and discussion regarding the free amino acid diets, especially TAA.
Lines 273–275: Although the text mentions a slight increase in hemocyte concentration in bees treated with TAA, the increase is so minimal that it might not be worth discussing.
R: Please note that after a reviewer suggested that we should discuss the hemocyte results, we concluded that instead of presenting a single graph averaging all results over a month, it would be better to present a graph of the hemocyte count per day (panel A) and a graph of the hemocyte count during the first week (panel B). The first graph shows an age-dependent decrease in Ubee's hemocyte count that was not previously appreciated. On the other hand, Panel B shows a period when all the analyzed groups could be compared. This new data analysis shows that Ubee has the highest hemocyte count. We believe that although there are still no significant differences between the diets, the difference between the sugar control and Ubee (p = 0.07) is worth mentioning.
Figure 3: The consumption of syrup, diet, and water is reported in grams, but the time unit is not specified.
R: Thanks for pointing out this mistake. It has been corrected.
Figure 4: The x-axis is labeled “Hemocitos” in Spanish instead of being translated into English.
R: Thank you. Corrected in the new figure 4.
The sentence “These findings have practical implications for beekeeping and physiological significance. First, they suggest that the combined use of both biomarkers could help characterize the nutritional value of different pollen substitutes more effectively” is redundant, as it is repeated in lines 343–345.
R: We were unable to find this sentence repeated.
The discussion starts by addressing the expression of mrjp1 and vg mRNAs, which does not follow the same order as the results. The first results presented pertain to survival percentage and food consumption, and this inconsistency can confuse readers.
R: We have organized the discussion topics according to the results. Additionally, we included a section for hemocytes as requested for another review (lines 389-394).
Round 2
Reviewer 3 Report
Comments and Suggestions for Authors
The manuscript has undergone significant improvements in response to the reviewers' comments. Given these substantial improvements, I recommend the publication of the article in its present form.